# Columbia Suicide Severity Rating Scale: Evidence of Construct Validity in Argentinians

**DOI:** 10.3390/bs13030198

**Published:** 2023-02-23

**Authors:** Pablo Ezequiel Flores-Kanter, Claudia Alesandrini, Jesús M. Alvarado

**Affiliations:** 1Consejo Nacional de Investigaciones Científicas y Técnicas (CONICET), Argentina & Centro de Bioética, Universidad Católica de Córdoba, Córdoba 5000, Argentina; 2Facultad de Psicología, Universidad Nacional de Córdoba, Córdoba 5000, Argentina; 3Department of Psychobiology & Behavioral Sciences Methods, Faculty of Psychology, Universidad Complutense de Madrid, Campus de Somosaguas S/N, 28223 Pozuelo de Alarcon, Spain

**Keywords:** C-SSRS, self-report, suicidality

## Abstract

Suicide is a global public health problem. The goal of this study was to evaluate the psychometric properties of the measurement of suicide severity based on the Columbia suicide severity rating scale. We worked with a sample of 516 Argentinean adults, aged 18 to 75. The fit of a measurement model that differentiates between the various degrees of suicidal severity was verified. The specified model returns fit values above the suggested cut-off points, both for the occurrence and frequency indicators. The internal consistency indices from the composite reliability coefficient also show values above the cut-off points for both occurrence and frequency. Finally, evidence of construct validity was obtained from the relationship with external variables. The results are consistent with the theory, showing stronger effects of hopelessness on suicidal ideation compared to suicide attempts. Overall, evidence of construct validity for the measurement of suicidal severity is presented, a contribution that is essential in remedying the lack of studies on suicide in the region and promoting prevention strategies.

## 1. Introduction

Suicide constitutes a public health problem that has a significant economic, social, and psychological impact globally [1,2,3]. In 2019, more than 700,000 cases of death by suicide were recorded worldwide. However, it is important to highlight that 77% of these suicide cases have been recorded in low- and middle-income countries [4]. The latest epidemiological data on Argentina [5] indicate that in 2020, a total of 3171 deaths by suicide were recorded, which is equivalent to a rate of 7.6 per 100,000 population. Of the recorded causes of violent death, suicide ranks as the main cause (35%). Therefore, while the suicide rate has fallen 12.1% compared to 2019, it is important that this public health problem continues to be addressed. Looking beyond the cases of death by suicide, the world mental health report [6] estimates that there are 20 suicide attempts per every death globally. All this has led various world academic, governmental, and non-governmental bodies to continue their calls for measures to be taken to prevent suicide [4,7].

A prerequisite for the prevention and treatment of the problem of suicide is for these prevention and intervention actions to be based on reliable scientific evidence [8]. Unfortunately, the accuracy and reliability of suicide statistics remain an issue to be solved in numerous countries [9]. To improve the quality of data linked to suicide, it is essential and necessary to have measurements that allow for valid inferences to be drawn on suicidal thoughts and behaviours [10]. One of the instruments developed for the purpose of providing valid measurements of suicidal thoughts and behaviour is the Columbia Suicide Severity Rating Scale (C-SSRS). The C-SSRS was designed to (i) offer definitions of suicidal ideation and behaviour, non-suicidal self-injurious behaviour, and corresponding assessments; (ii) quantify the full spectrum of suicidal ideation and suicidal behaviour and measure their severity over set periods; and (iii) distinguish between suicidal behaviour and non-suicidal self-injurious behaviour. These criteria are deemed indispensable in gauging the utility of suicide rating scales [11]. This study will address the construct validity of suicide severity measurement based on the application of the self-report version of the C-SSRS [12].

Globally, to date, we have identified few prior studies that have applied the self-report version of the C-SSRS [13,14,15,16], and only one prior study that has obtained evidence of construct validity based on the application of this type of version [12]. In their study, Viguera et al. [12] present evidence of validity in the use of cut-off points to establish the presence of suicide risk based on suicidal ideation and behaviour indicators. In comparison with the use of item 9 of the Patient Health Questionnaire-9, the C-SSRS showed greater sensitivity and specificity. In Argentina, although there is a specific prior example concerning the application and assessment of the psychometric properties that arise from applying the C-SSRS [17], some significant limitations are identified that merit a complementary study in this area. Firstly, it should be mentioned that the report is not clear regarding the format of the C-SSRS that has been applied. It is important to clarify at this point that different formats of application of the C-SSRS have been proposed, such as self-report versions, semi-structured interviews, or computerised versions (the different formats of the C-SSRS can be viewed at: https://cssrs.columbia.edu/, accessed on 15 September 2022). Judging by the contents of this study, it is likely that the Spanish version has been used, which is in the format of a semi-structured interview [18]. The limitations of this earlier study include (a) the use of a convenience sampling method with university students and the use of a specific region in Argentina (i.e., Rosario); (b) the application of non-recommended methods for factor analysis from a latent variable approach, as is the case of principal component analysis when applying exploratory factor analysis; and (c) not considering estimation methods suitable for the dichotomous and polytomous nature of many of the answer options in this scale. On this last point, it should be noted that the scale assesses two distinct facets of suicide severity and uses a specific response format for each one (i.e., it assesses the occurrence of indicators based on dichotomous responses and assesses the frequency with which these indicators have presented using a 5-point Likert scale). This aspect of relevance has not been considered in the earlier study. In light of the above, this project aims to overcome these limitations by (1) using a sample of Argentinian men and women from regions of the country that are more heterogenous and (2) using factorial methods that are more suited to the aims of the analysis and the characteristics of the data (i.e., polychoric and tetrachoric correlation matrices and appropriate estimation methods, such as weighted least squares).

Thus, the main goal of this study is to assess the psychometric properties derived from the self-report version of the C-SSRS. Specifically, it seeks to assess construct validity by assessing internal structure, an aspect which, to date, has not been assessed in the prior literature reviewed. Additionally, evidence of validity will also be obtained based on the relationship with the measurement of hopelessness (i.e., an external source of construct validity). It is held that the results of this project will be relevant in expanding global evidence of the validity of suicide severity ratings derived from the self-report version of the C-SSRS. Therefore, we aim to contribute to the campaign to meet suicide detection, prevention, treatment, and monitoring goals, specifically in Argentina and countries in the region.

## 2. Materials and Methods

### 2.1. Participants and Sampling Procedure

A total of 516 subjects participated in the survey. They were selected using a non-probability open-mode online sample method [19]. This data collection method has proven to be equivalent to traditional forms of collection (i.e., face-to-face) [20], returning similar results in terms of means, internal consistency indexes, correlations, response rates, and level of conformity with the questionnaire. The sample for this observational, cross-sectional study was collected using an online survey format to gather information through the Google Forms platform and was delivered by Facebook. The data were collected in September and October of 2022.

Ages ranged from 18 to 75 (M = 34.87, SD = 11.92). Of the total sample, most identified as female (79.46%) (see Table 1). Table 2 shows participants’ academic education and Table 3 shows the distribution by participants’ employment status and income. The vast majority have primary (99.42%) and secondary (97.48%) education, and almost half report having completed tertiary or university studies (48.64%). Finally, the majority indicate that they are dependent employed (39.92%) or self-employed (26.94%), with the most frequent income being 80K pesos or less (36.39%).

### 2.2. Data Collection Instruments

*Columbia Suicide Severity Rating Scale (C-SSRS).* The version applied herein is the format intended for the C-SSRS self-report screener [12,14], which employs the translation into Spanish by Al-Halabí et al. [18]. The instrument was developed with the aim of measuring two main dimensions according to the degree of suicidal severity: (a) suicidal ideation and (b) suicidal behaviour. The first dimension explores suicidal thoughts in a progressive manner (i.e., from lesser to greater severity) subdivided into passive suicidal ideation, with no specific plan or intent to act, and active suicidal ideation, with a plan and intent to act. The second dimension, suicidal behaviour, examines suicide attempts (i.e., action targeted at ending one’s life) and interrupted suicide attempts (i.e., action targeted at ending one’s life which is self-interrupted or interrupted by an external actor). Both dimensions probe both the occurrence (i.e., yes or no) and frequency (e.g., from less than once a week to many times per day) of the various suicide severity indicators. In total, the instrument comprised 14 items, 7 to measure occurrence and 7 to measure frequency. Whereas the occurrence indicators use a dichotomous response format, the frequency indicators use a format that has 5 response options (from 1: less than once a week to 5: many times per day) in addition to an open format allowing the respondent to freely indicate the frequency (the instrument is included in https://osf.io/7rscg/, accessed on 20 December 2022).

*Beck Hopelessness Scale (BHS)* [21]. The version validated in Argentina by Flores-Kanter et al. [22] was applied. The scale consists of 11 dichotomous response items (i.e., true, false), and is applied to assess negative expectations about the future. The evidence for Argentina suggests an essentially single-factor structure. In predictive terms, the factor score derived from this factor structure presents a high discrimination capacity between people with high and no suicidal ideation. This discrimination capacity is higher than that achieved using the total raw score [22].

### 2.3. Data Analysis Process

With regard to the data analysis procedure followed, the following methodological decisions were made:1-Due to the variability in items 6 and 7 on frequency and so as to use consistent criteria for the response format, these indicators were respecified prior to subsequent analysis. The original response format for items 6 and 7 is open and does not provide categorised response options. In this case, the question refers to indicating the number of times the person has made suicide attempts. Since the originally reported frequencies ranged from 0 to 15, the recoding consisted of requesting that values greater than 5 be taken with the value 5, while the rest of the values remained with the original value. So here, respecification consisted of recoding the responses with the range of 0 to 5, as is the case with the rest of the questions on frequency.2-Because the scale assesses two distinct facets of suicide severity and uses a specific response format for each one (i.e., it assesses the occurrence of indicators based on dichotomous responses and the frequency with which these indicators have presented using a 5-point Likert scale), we decided to carry out two separate factor analyses: a factor analysis for the occurrence indicators and a factor analysis for the frequency indicators.3-Measurement models for three correlated factors were specified, which examine suicide severity in ascending order. This model is consistent with the aim of the C-SSRS where the intention is to distinguish the domains of suicidal ideation and suicidal behaviour based on their severity degree [11]. In this sense, the elaboration and objectives followed by the C-SSRS are contrary to the traditional view that considers suicidal ideation and suicidal behaviour a unidimensional construct. In line with previous applications of the C-SSRS [12], the specified measurement models defined the factors based on the degree of severity. Thus, the following factors were specified in the measurement models: (1) passive suicidal ideation (i.e., no intent to act); (2) active suicidal ideation (i.e., intent to act); and (3) suicide attempt. Although it would be possible to specify a two-factor measurement model, separating only between suicidal ideation and suicidal behaviour, the C-SSRS has been constructed for the purpose of gradually differentiating different levels of suicidal severity. In this sense, we consider that the fact of not differentiating within the dimension of suicidal ideation between passive and active suicidal ideation detracts from the discriminative capacity of the measurements in terms of being able to distinguish between less severe ideation (i.e., passive suicidal ideation) and more severe ideation (i.e., active suicidal ideation).4-Regarding the structural model, the specified model considers the effect of the hopelessness factor on the suicide severity factors (i.e., passive suicidal ideation, active suicidal ideation, and suicide attempt). Here, we hypothesise that hopelessness will have a stronger relationship with suicidal ideation than suicide attempts. This hypothesis is based on the cognitive model of suicidal behaviour [22], where it is postulated that suicidal ideation is the strongest predictor of the suicidal act, whereas hopelessness is the strongest predictor of suicidal ideation.

In the data analysis, the data were specified entirely in the R Studio program [23]. The analysis consisted of applying structural equation modelling (SEM). This analysis set was applied so as to verify the fit of both measurement and structural models. In both cases, the WLSMV (weighted least squares with mean- and variance-adjusted standard errors) model was used, given that the indicators for the respective factors are categorical [24]. Model fit was assessed using comparative fit indexes (CFI), root mean square error of approximation (RMSEA), and standardised root mean square residual (SRMR). CFI values equal to or above 0.90 and 0.95 are deemed indicators of acceptable and optimum data fit, respectively, while RMSEA values below 0.8 and 0.5 indicate acceptable and optimum fit, respectively [25]. For SRMR, it is suggested that values below or equal to 0.8 are an indicator of acceptable fit [26]. The lavaan package was used for the above-mentioned purposes [27]. Finally, internal consistency was calculated according to the composite reliability index (also termed omega), using the measurement model estimated in SEM as direct input [28]. We used the semTools package here [29].

## 3. Results

### 3.1. Measurement Models: Source of Structural Evidence

The specified model returns fit values above the suggested cut-off points both for the occurrence indicators (χ^2^ = 17.14, df = 11, *p* = 0.10, CFI = 1.00, RMSEA = 0.03, SRMR = 0.05) and the frequency indicators (χ^2^ = 13.28, df = 11, *p* = 0.28, CFI = 1.00, RMSEA = 0.02, SRMR = 0.03).

The measurement models are laid out below with their respective standardised factor loadings (Figure 1 and Figure 2).

For both occurrence and frequency, all the factor loadings obtained reach statistical significance (*p* < 0.001), with these loadings ranging from 0.83 to 0.99 for occurrence and 0.85 to 0.95 for frequency (the standard errors for the estimate in the occurrence measurement model are in the range between 0.056 and 0.096, while in the frequency measurement model, it is in the range between 0.033 and 0.083).

Lastly, we present evidence of internal consistency. To do so, composite reliability was considered. The results showed values above the suggested cut-off points both for occurrence (CR _Passive Ideation_ = 0.80, CR _Active Ideation_ = 0.78, CR _Attempt_ = 0.76) and frequency (CR _Passive Ideation_ = 0.87, CR _Active Ideation_ = 0.79, CR _Attempt_ = 0.78).

### 3.2. Structural Models: Source of External Evidence

The hopelessness rating was considered here and the structural model was assessed, in which the effect of hopelessness on the passive suicidal ideation, active suicidal ideation, and attempt factors are specified (Figure 3 and Figure 4).

Both specified structural models obtain fit indicators above the specified cut-off points for both occurrence (χ^2^ = 178.75, df = 129, *p* = 0.002, CFI = 0.99, RMSEA = 0.03, SRMR = 0.08) and frequency (χ^2^ = 180.73, df = 129, *p* = 0.002, CFI = 0.99, RMSEA = 0.03, SRMR = 0.07). Additionally, all standardised regression weights are statistically significant (*p* < 0.001). The regression coefficients obtained for the structural part of the model range from 0.41 to 0.67 for the occurrence case, and from 0.41 to 0.66 for the frequency case. Finally, a stronger effect of hopelessness on the dimensions of suicidal ideation compared to the dimension of suicide attempt is observed. The latter result is similar for both suicidal occurrence (*β* _Passive Ideation_ = 0.64; *β* _Active Ideation_ = 0.62; *β* _Attempt_ = 0.41) and suicidal frequency (*β* _Passive Ideation_ = 0.66; *β* _Active Ideation_ = 0.62; *β* _Attempt_ = 0.41).

## 4. Discussion

The main goal of this study was to assess the psychometric properties derived from the self-report version of the C-SSRS. To do so, construct validity was assessed based on (a) a source of structural evidence and (b) a source of external evidence.

With regard to the source of structural evidence, it was verified that the measurement model for three correlated factors, which examines suicide severity on a rising scale, presents an optimal fit according to the commonly adopted criteria in the international literature. This measurement model screens the following factors: (1) passive suicidal ideation (i.e., no intent to act); (2) active suicidal ideation (i.e., intent to act); and (3) suicide attempt. Moreover, the specified factors and their respective items show evidence of internal consistency, measured using composite reliability, which was also above the commonly accepted cut-off points. These results are consistent with the original aim of the C-SSRS, which was designed to offer multidimensional measurement covering related but differing constructs [11], in an effort to distinguish the domains of suicidal ideation and suicidal behaviour.

With regard to external evidence, the hopelessness scale showed a stronger effect with measurements of suicidal ideation than those of suicide attempts. These results are consistent with the theoretical cognitive models of suicide, which posit hopelessness as the determining factor in suicidal thoughts [2]. In this model, suicidal ideation is the strongest predictor of suicidal behaviour [30].

These data are relevant both globally and regionally. Global relevance is due to the fact that they provide evidence that, to date, we have been unable to find on structural evidence or measurement model fit based on C-SSRS measurements, as well as structural model fit, including external evidence with a hopelessness measurement. Regionally, they are relevant because while in Argentina instruments are used to measure suicidal ideation (i.e., Patient Health Questionnaire-9) and suicide risk (i.e., ISO-30), these scales have a series of limitations, which means that using them is not advisable. In the case of Patient Health Questionnaire-9, it has been demonstrated that using item 9 as an indicator generated much higher rates of false-positive findings than the self-report version of the C-SSRS [12]. The ISO-30 suicide orientation scale attempts to offer a measurement for suicide risk. However, there is no evidence supporting the use of a general measurement (i.e., total score) of risk by means of ISO-30, and the constructs that have been theoretically taken into consideration in the creation of this scale also do not offer clear or stable structural evidence, except in the case of the suicidal ideation factor [31]. Therefore, this study offers an alternative measurement for use in the region, especially in Argentina, to obtain accurate and reliable data on suicide, by examining suicide severity on a rising scale and including measurements of suicidal ideation and suicidal behaviour.

This study presents some limitations that must be addressed in future applications of the scale. We note that (a) the sample size did not allow for analysis of invariance in the measurement and structural models, covering important variables such as the region in the country, gender, and age; (b) a general set value was considered in this study assessing the occurrence and frequency of suicidal ideations and behaviours, but other possibilities, such as the lifetime set value, were not included or compared; and (c) the stability of the measurement and structural models was not assessed in separate samples or over time. Finally, we would like to add a comment on the response format used by the C-SSRS. The C-SSRS asks about two aspects, the occurrence and frequency of indicators of suicidal severity. However, one of the reviewers has pertinently pointed out that the occurrence responses can easily be derived from the frequency of the responses; therefore, the number of questions in the C-SSRS could be reduced by half (i.e., by eliminating the questions referring to occurrence). Related to this last point, the reviewer also rightly noted that the results derived from the structural equation models are very similar for the occurrence and frequency indicators, which makes the added value of considering both indicators debatable. The latter can also be taken as evidence in favour of reducing the number of questions in the C-SSRS by considering only the frequency indicators. Of course, in the latter case, it would be relevant to incorporate a response option referring to “Never happened to me”. So as to continue securing evidence of construct validity for the measurements derived from the self-report version of the C-SSRS applied herein, it is important that future research pays attention to these limitations and be able to provide data on the matter.

## 5. Conclusions

This study provides evidence of construct validity for measurements taken using a self-report version of the C-SSRS. This proves relevant as the version used herein meets the criteria given by Roaten et al. [14] to inform the selection of suicide severity and risk scales (1) measurement validity/reliability, (2) brevity of application, (3) open access or availability in the public domain, and (4) administration with minimal training. Thus, C-SSRS, supported with timely clinical assessment, in conjunction with other measurement methods [10] (e.g., smartphone apps, wearable technology), may be a useful and efficient method of screening for suicide severity and suicide risk [14]. We hope that this report constitutes a key step in promoting more accurate and reliable data in the Central and South America region.

## Figures and Tables

**Figure 1 behavsci-13-00198-f001:**
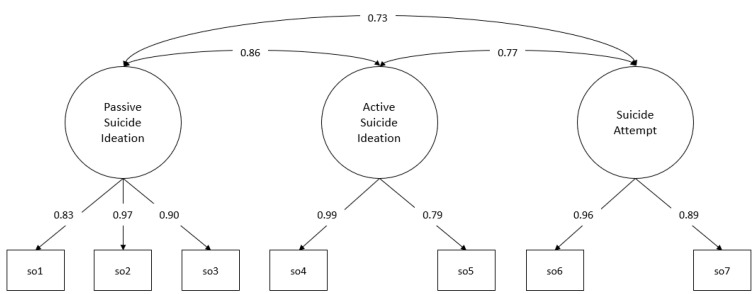
Measurement model: occurrence. so1–so7: suicide severity indicators, items 1 to 7. Standardised factor loadings and interfactor correlations.

**Figure 2 behavsci-13-00198-f002:**
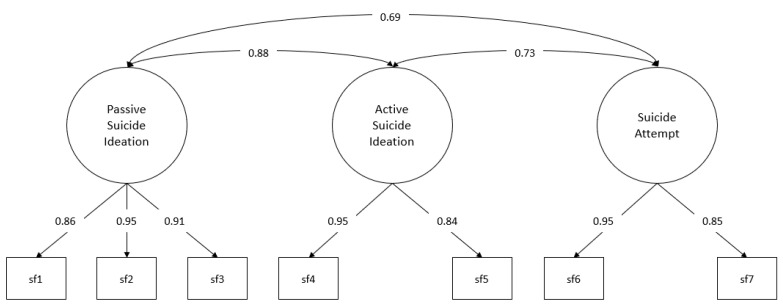
Measurement model: frequency. sf1–sf7: suicide severity indicators, items 1 to 7. Standardised factor loadings and interfactor correlations.

**Figure 3 behavsci-13-00198-f003:**
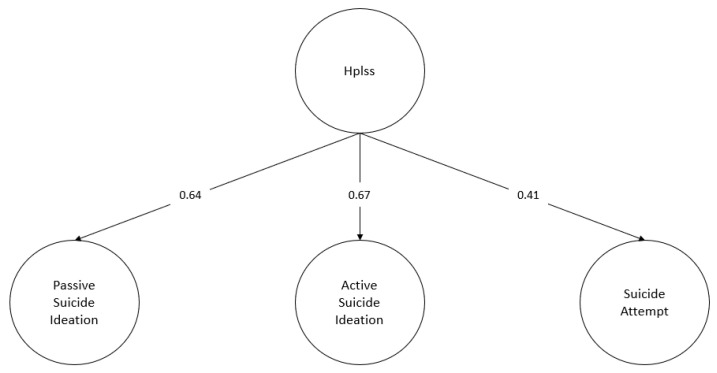
Structural model: occurrence. Hplss: hopelessness. Standardised regression coefficients. For the sake of clarity, observable indicators for each latent factor considered in the model are omitted from the diagram. Likewise, the interfactor correlations between the suicide severity factors were omitted.

**Figure 4 behavsci-13-00198-f004:**
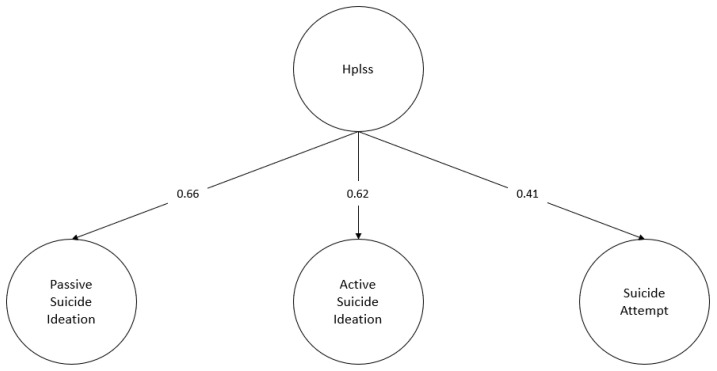
Structural model: frequency. Hplss: hopelessness. Standardised regression coefficients. For the sake of clarity, observable indicators for each latent factor considered in the model are omitted from the diagram. Likewise, the interfactor correlations between the suicide severity factors were omitted.

**Table 1 behavsci-13-00198-t001:** Sociodemographic descriptors.

Gender	*n*	%	Pyscho_treat	*n*	%	Pyschi_treat	*n*	%
Female	410	79.46%	Yes	142	27.52%	Yes	44	8.53%
Male	101	19.57%	No	374	72.48%	No	472	91.47%
Non-binary	5	0.97%						

Note. *n* = size; % = percentage; Pyscho_treat = psychological treatment; Pyschi_treat = psychiatric treatment.

**Table 2 behavsci-13-00198-t002:** Academic education descriptors.

Primary	*n*	%	Secondary	*n*	%	T/U	*n*	%
Yes	513	99.42%	Yes	503	97.48%	Yes	251	48.64%
No	3	0.58%	No	13	2.52%	No	265	51.36%

Note. *n* = size; % = percentage; T/U = tertiary or university education. Corresponding, respectively, to elementary, middle, and high school in the US.

**Table 3 behavsci-13-00198-t003:** Employment status and income descriptors.

Employment Status	*n*	%	Income	*n*	%
Homemaker	10	1.94%	<40k	177	34.3%
Self-employed	139	26.94%	40–80k	167	32.36%
Unemployed	49	9.5%	80–95k	59	11.43%
Student	86	16.67%	95–120k	48	9.3%
Retired or pensioner	26	5.04%	120–380k	61	11.82%
Dependent	206	39.92%	>380k	4	0.78%

Note. *n* = size; % = percentage. The income is indicated in Argentine currency. At the time of writing this part of the report (14 February 2023), 1 USD corresponds to approximately 380 ARS.

## Data Availability

Data supporting the reported results can be downloaded at https://osf.io/7rscg/ (accessed on 20 December 2022). The files are hosted in an Open Science Framework (OSF) repository and can be accessed at any time.

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
