# Peer review of "Columbia Suicide Severity Rating Scale: Evidence of Construct Validity in Argentinians"

_behavsci, 2023, doi:10.3390/bs13030198_

Round 1
Reviewer 1 Report
This study aimed to evaluate the construct validity of the Columbia-suicide severity rating scale in Argentinians, the results showed stronger effects of hopelessness on suicidal ideation compared to suicide at-18 tempts, which presented evidence of construct validity for a measurement of suicidal severity in this area. Besides, the spelling need to be checked, eg. in Line 73 'on the one hand'
Author Response
Response: Thank you very much. We have checked again the spelling. With regard to line 73, the wording of the sentence has been changed:
“On this last point, it should be noted that the scale assesses two distinct facets of suicide severity and uses a specific response format for each one (i.e., it assesses the occurrence of indicators based on dichotomous responses and assesses the frequency with which these indicators have presented using a 5-point Likert scale).”
Reviewer 2 Report
Review is provided in separate pdf file

Author Response
1- Some sentences seem unnecessarily complicated, for example: “Firstly, it should be mentioned that the report is not fully transparent on the data collection procedure it follows and the C-SSRS format it applies, which makes it difficult to ascertain which specific version of the scale has been applied (i.e. self-report vs semi-structured clinical interview).” Up to this point it has not been mentioned that the C-SSRS can be used in different formats/versions? If semi-structured interview is one format, which are the others?
Response: Thank you very much. We have amended the relevant paragraph for clarity:
“Firstly, it should be mentioned that the report is not clear regarding the format of the C-SSRS that has been applied. It is important to clarify at this point that different formats of application of the C-SSRS have been proposed, such as self-report versions, semi-structured interviews, or computerised versions.”
Please also see the footnote incorporated therein.
2- The sample consists of 516 persons. I would have thought that suicidal ideation occurs and suicidal attempts occur only in selective groups (e.g. persons with depressive symptoms). Is a random-like sample suitable for construct validity investigation, if, for example, only a small fraction of examinees has had depressive experience?
Response: In our sample, the frequency of the dimensions of suicidal severity varies according to the degree of severity. We observe, for example, that in the case of dimensions of lower suicidal severity, such as passive suicidal ideation, the frequency varies between 22.09% and 36.24%. Like any other analysis based on statistical association between variables, what is relevant for the type of factor analysis applied here is that there is a range of variability in the measurement taken as input. In summary, although the frequency of the behaviours measured may be low, they still present a variability such that the factor analyses proposed here can be applied.
3- The description of the C-SSRS in section 2.2 is little confusing. Authors wrote: “The instrument considered two categories in the suicide severity assessment: a) suicidal ideation and b) suicidal behaviour.” The models in 3.1, however, represent latent dimension/factors rather than categories.
Response: Thank you very much. We have amended the relevant paragraph for clarity. Please see the modifications made in 2.2. Data Collection Instruments “Columbia Suicide Severity Rating Scale (C-SSRS). “
4- Additionally, these models operationalize three factors instead of two. More-over, authors state: “it was verified that the measurement model for three correlated factors, which examines suicide severity in a rising scale […]”. In my understanding, a “rising scale” should be something like an uni-dimensional construct—for example if I separate a continuous scale into two areas by setting a cut score. In 3.1.1, authors state that “in contrast to the traditional view that considers suicidal ideation and suicidal behaviour a one-dimensional construct (i.e. a continuum of passive ideation, active intent and behaviour), the intention is to distinguish the domains of suicidal ideation and suicidal behaviour”. These two controversial view should be contrasted much earlier (in the introduction), because otherwise, readers are confused why a three-dimensional model is defined for an instrument with two categories which was measured unidimensional in the past.
Response: Thank you very much. We have incorporated information in the relevant section for clarity:
“With regard to the data analysis procedure followed, the following methodological decisions were made:
1- Due to the variability in items 6 and 7 on frequency and so as to use consistent criteria for the response format, these indicators were respecified prior to subsequent analysis (the original response format for items 6 and 7 is open and does not provide categorised response options). Respecification consisted of recoding the responses with the range of 0 to 5, as is the case with the rest of the questions on frequency.
2- Because the scale assesses two distinct facets of suicide risk and uses a specific response format for each one (i.e., it assesses the occurrence of indicators based on dichotomous responses and the frequency with which these indicators have presented using a 5-point Likert scale), we decided to carry out two separate factor analyses: a factor analysis for the occurrence indicators and a factor analysis for the frequency indicators..
3- A measurement model for three correlated factors was specified, which examines suicide severity in ascending order. This model is consistent with the aim of the C-SSRS where, in contrast to the traditional view that considers suicidal ideation and suicidal behaviour a unidimensional construct, the intention is to distinguish the domains of suicidal ideation and suicidal behaviour based on their severity degree [11]. In function on this and in line with previous applications of the C-SSRS [12], the specified measurement model defined the factors based on degree of severity. Thus, the following factors were specified in the measurement model: 1) Passive suicidal ideation (i.e. no intent to act); 2) Active suicidal ideation (i.e. intent to act); and 3) Suicide attempt. Although it would be possible to specify a two-factor measurement model, separating only between suicidal ideation and suicidal behaviour, the C-SSRS has been constructed for the purpose of gradually differentiating different levels of suicidal severity. In this sense, we consider that the fact of not differentiating within the dimension of suicidal ideation between passive and active suicidal ideation detracts from the discriminative capacity of the measurements in terms of being able to distinguish between less severe ideation (i.e., passive suicidal ideation) and more severe ideation (i.e., active suicidal ideation).
4- Regarding the structural model, the specified model considers the effect of the hopelessness factor on the suicide severity factors (i.e., passive suicidal ideation; active suicidal ideation; suicide attempt). Here we hypothesize that hopelessness will have a stronger relationship with suicidal ideation than suicide attempts. This hypothesis is based on the cognitive model of suicidal behaviour [22] where it is postulated that suicidal ideation is the strongest predictor of the suicidal act, whereas hopelessness is the strongest predictor of suicidal ideation.”
5- Similarly, authors state in the discussion that their “results are consistent with the theoretical cognitive models of suicide, which posit hopelessness as the determining factor in suicidal thoughts. In this model, suicidal ideation is the strongest predictor of suicidal behavior.” To draw this conclusion, the autors would have had to specify a regression model instead of a latent correlation model.
Response: Thank you. We have clarified this point better in:
“
4- Regarding the structural model, the specified model considers the effect of the hopelessness factor on the suicide severity factors (i.e., passive suicidal ideation; active suicidal ideation; suicide attempt). Here we hypothesize that hopelessness will have a stronger relationship with suicidal ideation than suicide attempts. This hypothesis is based on the cognitive model of suicidal behaviour [22] where it is postulated that suicidal ideation is the strongest predictor of the suicidal act, whereas hopelessness is the strongest predictor of suicidal ideation.”
Then we mentioned: “On the data analysis, the data were specified entirely in the R-Studio programme [23]. The analysis consisted of applying structural equation modelling (SEM). This analysis set was applied so as to verify the fit of both measurement and structural models.”
Hopefully this makes it clearer that two models are specified from SEM, a measurement model (for evidence of internal structure) and a structural model (for evidence of test-criteria).
6- The factor loadings in the figures (between the green arrows) are hard to see
Response: Thank you very much. We have modified the corresponding figures.
7- Authors state in the introduction: “One the one hand, it assesses the occurrence of indicators based on dichotomous responses and, on the other hand, assesses the frequency with which these indicators have presented using a 5-point Likert scale. This aspect of relevance has not
been considered in the earlier study.” According to this distinction: which measurement model was used? A linear SEM for frequency and a logit/probit model for dichotomous responses of occurrence?
Response: We have specified this in the data analysis part, where we have indicated that:
“…the WLSMV (weighted least squares with mean- and variance-adjusted standard errors) model was used, given that the indicators for the respective factors are categorical [24].“
As indicated in the introduction, the use of this factorial approach “… are more suited to the aims of the analysis and the characteristics of the data (i.e. polychoric and tetrachoric correlation matrices and appropriate estimation methods, such as weighted least squares).”
We hope that the changes have made this approach clearer.
8- Section 3.1.1 might be enhanced. Authors might introduce that first, the three-dimensional measurement model is specified separately for occurrence and frequency, and afterwards, the latent structure between these three dimensions and hopelessness is investigated. Authors might repeatedly mention that first step is to verify internal consistency, and second step is for construct validity. Authors should state what they expect if construct validity holds (and why). To me, it is not completely clear why hopelessness should have a stronger relationship with suicidal ideation than with suicide attempts.
Response: To clarify these points, we have made modifications in section 2.3. Please see the additions referenced in response to the above comment (number 4).
In addition, we have incorporated information in section 3.1:
“For both occurrence and frequency, all the factor loadings obtained reach statistical significance (p < .001), with these loadings ranging from .83 to .99 for occurrence and .85 to .95 for frequency (the standard errors for the estimate in the occurrence measurement model are in the range between .056 and .096, while in the frequency measurement model it is in the range between .033 and .083).”
And later in the same section:
“Both specified structural models obtain fit indicators above the specified cut-off points for both occurrence (χ2= 178.75, df = 129, p = 0.002, CFI = .99, RMSEA = 0.03, SRMR = 0.08) and frequency (χ2= 180.73, df = 129, p = 0.002, CFI = .99, RMSEA = 0.03, SRMR = 0.07). Also, all standardised regression weights are statistically significant (p < .001). The regression coefficients obtained for the structural part of the model range from .41 to .67 for the occurrence case, and from .41 to .66 for the frequency case. Finally, a stronger effect of hopelessness on the dimensions of suicidal ideation compared to the dimension of suicide attempt is observed. The latter result is similar for both suicidal occurrence (β Passive Ideation = .64; β Active Ideation = .62; β Attempt = .41) and suicidal frequency (β Passive Ideation = .66; β Active Ideation = .62; β Attempt = .41).”
9- Description of figure 3 and 4 is in Spanish
Response: Thank you very much. Corrected.
10- Figure 3 and 4: Please explain what bold/thin arrows, light and dark green, and solid/dashed lines stand for
Response: Thank you very much. We have modified the corresponding figures.
11- Please explain shortly composite reliability measures as CRP_I, CRA_I, CRAtt
Response: Thank you very much. We have corrected the paragraph as follows:
“Lastly, we present evidence of internal consistency. To do so, composite reliability was considered. The results showed values above the suggested cut-off points both for occurrence (CR Passive Ideation = 0.80, CR Active Ideation = 0.78, CR Attempt = 0.76) and frequency (CR Passive Ideation = 0.87, CR Active Ideation = 0.79, CR Attempt = 0.78).”
12- From my point of view it would be clearer to combine table 1, 2 and 3 into one table, and to rearrange as follows:
Response: While a good suggestion, it is visually more cumbersome when trying to fit it into the journal format. This is why we have decided to leave the tables as they are.
Reviewer 3 Report
This article deals with the study of the psychometric properties of a brief scale to assess the severity of suicide in an Argentine sample. Specifically, evidence is obtained of its construct validity, as well as its external validity. The manuscript is very well written and organized, the analyzes are adequately described, and the results are clearly presented. I would just like to make a few points.
In the procedure section, when the authors describe the data collection, I think they have forgotten to indicate how it was carried out. It is indicated that this method (I assume it is online) has been shown to be equivalent to the traditional way of collecting data face to face.
On the other hand, when the measurement instrument is described, it is mentioned that it consists of seven items that measure occurrence (dichotomous response scale), and another seven that measure frequency (five-option response scale). However, neither in this section nor in the analysis section is it clear whether both subscales are factored together or separately, and in that case, whether each subscale is supposed to measure three factors. The reader can only know when he reaches the Results section. I think it should be specified in one of the previous sections.
Regarding the figures that are offered, the factorial saturation of the items can hardly be seen. I guess they are the ones offered by R. I think it would be convenient to do them separately, in PowerPoint it is easy and fast. Or the factor loadings obtained could be commented on in the text, as well as whether they are statistically significant and offer the value of p. Due to their size, they must be, but I think it doesn't hurt to point it out. And regarding the value of these factor loadings, I observe that some are very high (.95 or .96). Can they really be considered significantly different from 1? I think that the confidence interval could be established around those values, since if 1 is included then perhaps this three-dimensional structure cannot be sustained. I would also like to know what the lines in the figures mean, since lines of different types are offered, and it does not seem that the thickest lines correspond to the items with the most weight, necessarily. Finally, the headings of figures 3 and 4 are written in Spanish, they must be translated into English...
Author Response
1- In the procedure section, when the authors describe the data collection, I think they have forgotten to indicate how it was carried out. It is indicated that this method (I assume it is online) has been shown to be equivalent to the traditional way of collecting data face to face.
Response: Thank you very much. We have amended the relevant paragraph for clarity:
“A total of 516 subjects participated in the survey. They were selected using a non-probability open mode online sample method [19]. This data collection method has proven to be equivalent to traditional forms of collection (i.e. face-to face) [20], returning similar results in terms of means, internal consistency indexes, correlations, response rates, and level of conformity with the questionnaire. The sample fort his observational, cross‐sectional study was collected using an online survey format to gather information through the Google Forms platform and was delivered by Facebook social media. The data were collected in September and October of 2022.”
2- On the other hand, when the measurement instrument is described, it is mentioned that it consists of seven items that measure occurrence (dichotomous response scale), and another seven that measure frequency (five-option response scale). However, neither in this section nor in the analysis section is it clear whether both subscales are factored together or separately, and in that case, whether each subscale is supposed to measure three factors. The reader can only know when he reaches the Results section. I think it should be specified in one of the previous sections.
Response: Thank you very much. We have incorporated information in the relevant section for clarity:
“With regard to the data analysis procedure followed, the following methodological decisions were made:
1- Due to the variability in items 6 and 7 on frequency and so as to use consistent criteria for the response format, these indicators were respecified prior to subsequent analysis (the original response format for items 6 and 7 is open and does not provide categorised response options). Respecification consisted of recoding the responses with the range of 0 to 5, as is the case with the rest of the questions on frequency.
2- Because the scale assesses two distinct facets of suicide risk and uses a specific response format for each one (i.e., it assesses the occurrence of indicators based on dichotomous responses and the frequency with which these indicators have presented using a 5-point Likert scale), we decided to carry out two separate factor analyses: a factor analysis for the occurrence indicators and a factor analysis for the frequency indicators..
3- A measurement model for three correlated factors was specified, which examines suicide severity in ascending order. This model is consistent with the aim of the C-SSRS where, in contrast to the traditional view that considers suicidal ideation and suicidal behaviour a unidimensional construct, the intention is to distinguish the domains of suicidal ideation and suicidal behaviour based on their severity degree [11]. In function on this and in line with previous applications of the C-SSRS [12], the specified measurement model defined the factors based on degree of severity. Thus, the following factors were specified in the measurement model: 1) Passive suicidal ideation (i.e. no intent to act); 2) Active suicidal ideation (i.e. intent to act); and 3) Suicide attempt. Although it would be possible to specify a two-factor measurement model, separating only between suicidal ideation and suicidal behaviour, the C-SSRS has been constructed for the purpose of gradually differentiating different levels of suicidal severity. In this sense, we consider that the fact of not differentiating within the dimension of suicidal ideation between passive and active suicidal ideation detracts from the discriminative capacity of the measurements in terms of being able to distinguish between less severe ideation (i.e., passive suicidal ideation) and more severe ideation (i.e., active suicidal ideation).
4- Regarding the structural model, the specified model considers the effect of the hopelessness factor on the suicide severity factors (i.e., passive suicidal ideation; active suicidal ideation; suicide attempt). Here we hypothesize that hopelessness will have a stronger relationship with suicidal ideation than suicide attempts. This hypothesis is based on the cognitive model of suicidal behaviour [22] where it is postulated that suicidal ideation is the strongest predictor of the suicidal act, whereas hopelessness is the strongest predictor of suicidal ideation.”
3- Regarding the figures that are offered, the factorial saturation of the items can hardly be seen. I guess they are the ones offered by R. I think it would be convenient to do them separately, in PowerPoint it is easy and fast.
I would also like to know what the lines in the figures mean, since lines of different types are offered, and it does not seem that the thickest lines correspond to the items with the most weight, necessarily. Finally, the headings of figures 3 and 4 are written in Spanish, they must be translated into English...
Response: Thank you very much. We have modified the corresponding figures in response to your comments.
4- Or the factor loadings obtained could be commented on in the text, as well as whether they are statistically significant and offer the value of p. Due to their size, they must be, but I think it doesn't hurt to point it out.
Response: Thank you very much. We have incorporated information in the relevant section for clarity:
“For both occurrence and frequency, all the factor loadings obtained reach statistical significance (p < .001), with these loadings ranging from .83 to .99 for occurrence and .85 to .95 for frequency (the standard errors for the estimate in the occurrence measurement model are in the range between .056 and .096, while in the frequency measurement model it is in the range between .033 and .083).”
5- And regarding the value of these factor loadings, I observe that some are very high (.95 or .96). Can they really be considered significantly different from 1? I think that the confidence interval could be established around those values, since if 1 is included then perhaps this three-dimensional structure cannot be sustained.
Response: We have checked the standard error of the estimate of the measurement model and we have been able to verify that the values obtained are not equal to 0 (the standard errors for the estimate in the occurrence measurement model are in the range between .056 and .096, while in the frequency measurement model it is in the range between .033 and .083). We have incorporated this information into the manuscript:
“For both occurrence and frequency, all the factor loadings obtained reach statistical significance (p < .001), with these loadings ranging from .83 to .99 for occurrence and .85 to .95 for frequency (the standard errors for the estimate in the occurrence measurement model are in the range between .056 and .096, while in the frequency measurement model it is in the range between .033 and .083).”
Round 2
Reviewer 2 Report
Thanks for the revision. I think the manuscript improved substantially, and I have only some minor issues:
Line 101: typo ("The sample fort his observational" -> "for this observational")
Line 144-148: It might be cumbersome, but for the sake of comprehensibility, it should be described in more detail how these open responses were recoded into response categories (maybe in an attachment or supplement?).
Line 149-154: Thanks for clarification, but in my understanding, frequency and occurrence are closely related to each other, as frequency can be recoded into occurrence, can't it? For example, if I measure frequency with a 5-point Likert scale with 1 = never, 2 = seldom, ... and 5 = often, the occurrence might result from recoding frequency with 1 = no occurrence at all, and 2-5 = [some] occurrences. It could be that my problem relates more to the C-SSRS than how you modeled it, but it's hard for me to understand: Why bother participants with asking them about occurrence when you've already asked them about frequency? The results of Figure 1 and 2 are very similar - the same is true for figure 3 and 4. (perhaps you might add some annotations about that in the discussion?)
Line 155ff, suggested wording: Two measurement models for three correlated factors each were specified. Because the C-SSRS operationalizes suicidal ideation and suicidal behavior as two separate constructs, we modeled the two constructs as two correlated latent dimensions.
Language: I'm not a native speaker. Overall, the article is easy to understand, but from my perspective it is somewhat intricately worded in a few places.
Author Response
Thanks for the revision. I think the manuscript improved substantially, and I have only some minor issues:
Line 101: typo ("The sample fort his observational" -> "for this observational")
Response: Thank you, we have corrected that term.
Line 144-148: It might be cumbersome, but for the sake of comprehensibility, it should be described in more detail how these open responses were recoded into response categories (maybe in an attachment or supplement?).
Response: Thank you for your suggestion. We have modified part of the paragraph in the light of this suggestion:
“The original response format for items 6 and 7 is open and does not provide categorised response options. In this case, the question refers to indicating the number of times the person has made suicide attempts. Since the originally reported frequencies ranged from 0 to 15, the recoding consisted of requesting that values greater than 5 be taken with the value 5, while the rest of the values remained with the original value. So here, respecification consisted of recoding the responses with the range of 0 to 5, as is the case with the rest of the questions on frequency.”
Line 149-154: Thanks for clarification, but in my understanding, frequency and occurrence are closely related to each other, as frequency can be recoded into occurrence, can't it? For example, if I measure frequency with a 5-point Likert scale with 1 = never, 2 = seldom, ... and 5 = often, the occurrence might result from recoding frequency with 1 = no occurrence at all, and 2-5 = [some] occurrences. It could be that my problem relates more to the C-SSRS than how you modeled it, but it's hard for me to understand: Why bother participants with asking them about occurrence when you've already asked them about frequency? The results of Figure 1 and 2 are very similar - the same is true for figure 3 and 4. (perhaps you might add some annotations about that in the discussion?)
Response: Thank you for your suggestion. In light of this, we have incorporated the following sentence into the limitations of the discussion:
“Finally, we would like to add a comment on the response format used by the C-SSRS. The C-SSRS asks about two aspects, occurrence and frequency of indicators of suicidal severity. However, one of the reviewers has pertinently pointed out that the occurrence responses can easily be derived from the frequency responses; therefore, the number of questions in the C-SSRS could be reduced by half (i.e., by eliminating the questions referring to occurrence). Related to this last point, the reviewer also rightly noted that the results derived from the structural equation models are very similar for the occurrence and frequency indicators, which makes the added value of considering both indicators debatable. The latter can also be taken as evidence in favour of reducing the number of questions in the C-SSRS by considering only the frequency indicators. Of course, in the latter case it would be relevant to incorporate a response option referring to "Never happened to me".”
Line 155ff, suggested wording: Two measurement models for three correlated factors each were specified. Because the C-SSRS operationalizes suicidal ideation and suicidal behavior as two separate constructs, we modeled the two constructs as two correlated latent dimensions.
Response: Thank you for your suggestion. We have modified part of the paragraph in the light of this suggestion:
“Measurement models for three correlated factors were specified, which examines suicide severity in ascending order. This model is consistent with the aim of the C-SSRS where the intention is to distinguish the domains of suicidal ideation and suicidal behaviour based on their severity degree [11]. In this sense, the elaboration and objectives followed by the C-SSRS is contrary to the traditional view that considers suicidal ideation and suicidal behaviour a unidimensional construct. In function on this and in line with previous applications of the C-SSRS [12], here the specified measurement models defined the factors based on degree of severity. Thus, the following factors were specified in the measurement models:”
Language: I'm not a native speaker. Overall, the article is easy to understand, but from my perspective it is somewhat intricately worded in a few places.
Response: Thank you, we have revised the manuscript in an effort to achieve more fluent English and less intricately worded.